# Customizing Nutrients: Vitamin D and Iron Deficiencies in Overweight and Obese Children—Insights from a Romanian Study

**DOI:** 10.3390/nu17071193

**Published:** 2025-03-29

**Authors:** Raluca Maria Vlad, Oana-Andreea Istrate-Grigore, Daniela Pacurar

**Affiliations:** 1Department of Pediatrics, “Carol Davila” University of Medicine and Pharmacy, 020021 Bucharest, Romania; raluca.vlad@umfcd.ro (R.M.V.); daniela.pacurar@umfcd.ro (D.P.); 2“Grigore Alexandrescu” Emergency Children’s Hospital, Bld. Iancu de Hunedoara 30-32 Bucharest, 011743 Bucharest, Romania

**Keywords:** overweight children, childhood obesity, vitamin D insufficiency, iron deficiency

## Abstract

**Background**: Childhood obesity is a public health issue worldwide, recognized as a complex condition associated with multiple deficiencies in nutrients, such as vitamin D deficiency, iron-deficiency anemia, or abnormalities in serum calcium or phosphorus levels, despite an excess caloric intake. **Objective**: This study aims to investigate the prevalence of these deficiencies in overweight/obese children and to assess the correlation between vitamin D/serum iron levels and body mass index (BMI). **Methods**: The observational study was conducted on 69 Romanian overweight and obese children, aged 2 to 17, admitted to the Pediatrics Department of “Grigore Alexandrescu” Hospital in Bucharest over a 15-month period. The age- and gender-specific BMI percentiles were used to classify participants into three groups: overweight (≥85th and <95th BMI percentile), obese (≥95th BMI percentile), and severely obese (>120% of 95th BMI percentile). Data analysis focused on identifying the need for screening and targeted treatment in this pediatric population. **Results**: Hypovitaminosis D (defined as a serum level of 25(OH)D < 30 ng/mL) prevalence was significantly higher in the severe obesity category (71.5%) compared to the obesity (69%) and overweight (61.5%) groups. Iron deficiency and iron-deficiency anemia were both more prevalent in overweight children, with rates of 50% and 38.5%, respectively. Negative moderate correlations were found both between serum 25-hydroxyvitamin D levels and children’s age (r = −0.444, *p*-value < 0.0001), as well as between serum 25(OH)D levels and BMI (r = −0.31, *p* = 0.015), with no statistically significant correlation between serum iron level and BMI in this cohort (r = −0.02, *p* > 0.05). **Conclusions**: Severe obesity could be regarded as an associated factor for vitamin D insufficiency as this is highly prevalent in severely obese children, with 25(OH)D levels decreasing with the increase in BMI. Overweight children demonstrated an increased prevalence of iron deficiency in the overweight category, suggesting that the adipose tissue contributes to chronic inflammation, disrupting iron homeostasis. Given the high prevalence of nutritional deficiencies in this population, implementing systematic screening and treatment programs would be beneficial to prevent long-term adverse outcomes.

## 1. Introduction

Excessive body weight and obesity in children are escalating global health concerns, with their prevalence rising significantly in recent years [1]. Obese children are vulnerable to significant health risks, including metabolic and cardiovascular complications, like hypertension, dyslipidemia, and type 2 diabetes [2,3,4,5,6]. Among these complications, nutritional deficiencies have gained increasing attention due to an unbalanced diet characterized by the consumption of calorie-dense but nutrient-poor foods [2,3]. Studies indicate that these nutritional deficiencies may include vitamin D deficiency, iron-deficiency anemia, or abnormalities in calcium or phosphorus levels, despite excessive caloric intake [7,8].

Vitamin D is essential for skeletal health, calcium homeostasis, and immune function in developing children, with 25-hydroxyvitamin D (25(OH)D) being the primary circulating form used to assess vitamin D status [7,9].

Obesity is a recognized risk factor for vitamin D deficiency, indicating that adipose tissue may significantly contribute to low serum vitamin D levels, depending on geographic region, sun exposure, dietary habits, and methods of assessment [10,11].

Research has described several mechanisms to explain the relationship between obesity and hypovitaminosis D in children [12]. First, many behavioral factors contribute to hypovitaminosis D in obese children, including lower dietary consumption of vitamin D-enriched foods and decreased sun exposure due to limited outdoor activities. Additionally, the lipophilic nature of 25(OH)D causes its sequestration in adipose tissue, leading to inadequate circulating 25-hydroxyvitamin D (25(OH)D) concentrations [10]. Other mechanisms may contribute, such as a larger volume of distribution or impaired intestinal absorption [10].

The interplay between vitamin D insufficiency and fat accumulation creates a vicious cycle. This enzyme deficiency, caused by hepatic fat infiltration, results in the accumulation of inactive vitamin D forms and lowers its bioavailability [13].

Although the association between obesity and insufficient levels of vitamin D is described, constant research is needed to understand the long-term effects of correcting vitamin D insufficiency on obesity-related complications and to explore genetic or other environmental factors that may influence this relationship [14].

Iron-deficiency anemia is the most prevalent nutritional disorder worldwide and is linked to a higher risk of behavioral issues, learning difficulties, and cognitive delays in children [15,16].

Overweight and obesity, both considered low-grade inflammatory conditions, may disrupt iron homeostasis, leading to iron deficiency and iron-deficiency anemia [17,18]. Research has shown that obese children are nearly twice as likely to develop iron deficiency, with iron-deficiency anemia being significantly more prevalent in obese individuals compared to those with normal weight [19,20].

The association may be explained by obesity-related chronic inflammation. This inflammatory status stimulates the liver to produce hepcidin, a hormone that regulates iron metabolism. Elevated hepcidin levels inhibit iron absorption in the intestine and block iron release from macrophages and liver stores, leading to functional iron deficiency [21]. Low-grade inflammation in children who are overweight and obese can lead to elevated ferritin levels, as ferritin acts as an acute-phase reactant. This can mask true iron deficiency, making serum ferritin an unreliable marker of iron stores in overweight and obese children, delaying diagnosis and appropriate treatment [22].

Studies have shown that the inadequate consumption of bioavailable iron plays a significant role in the development of iron-deficiency anemia in overweight children, particularly those whose diets are primarily made up of processed and fast foods [19].

Overweight and obese children often experience rapid periods of linear growth, increasing their iron requirements to support the growth of red blood cell mass. If their dietary intake fails to meet these increased requirements, then they face an increased risk of developing iron deficiency or iron-deficiency anemia [20].

Due to intestinal inflammation and the consumption of processed foods, children who are overweight and obese are at higher risk of developing dysbiosis, which has been implicated in the reduced absorption of dietary iron due [23].

## 2. Materials and Methods

### 2.1. Study Cohort

We conducted a retrospective observational study over a 15-month period from September 2023 to November 2024 including 69 overweight/obese children admitted to “Grigore Alexandrescu” Emergency Clinical Hospital for Children in Bucharest, Romania, for conditions unrelated to obesity. The inclusion criteria consisted of ages between 2 and 17 years, the status of being overweight or obese (based on age- and gender-specific BMI percentiles provided by Center for Disease Control), and the availability of required data. The exclusion criteria included genetic syndromes (Down Syndrome, Prader–Willi Syndrome, Bardet–Biedl Syndrome, Beckwith–Wiedemann Syndrome) or inborn errors of metabolism (excluded using data in the patients’ medical records). Other exclusion criteria were patients with incomplete data. Patients underwent clinical examination and blood testing in the pediatric department.

### 2.2. Samples Collection

Weight (kg) and height (cm) measurements were recorded, and body mass index (BMI) was calculated for each patient. Based on age- and gender-specific BMI percentiles provided by the Centers for Disease Control (CDC2022), children were classified into three groups: overweight (≥85th and <95th BMI percentile), obese (≥95th BMI percentile), and severely obese (>120% of 95th BMI percentile) [24,25].

Data collection included a dietary survey from the medical charts of the included patients. Due to the retrospective design of the study, this information was limited, but dietary errors were identified in all children, including: the consumption of processed foods, a large number of snacks between main meals, and the consumption of calorie-dense but nutrient-poor foods.

Blood samples were collected from all patients by venipuncture in a fasting state (for 8 to 12 h of fasting) since they were all admitted to the hospital. We measured complete blood count with a focus on hemoglobin level (g/dL), mean corpuscular volume (MCV, fL), and mean corpuscular hemoglobin (MHC, pg). The analysis also included serum 25-hydroxyvitamin D (25(OH)D) level (ng/mL), serum iron level (mcg/dL), and calcium and phosphorus plasma levels.

The primary variable of interest, 25-hydroxyvitamin D, was determined by a high-specificity immunochemical method with detection by electrochemiluminescence with intra- and inter-assay coefficients of variation of 4.5% and 7.9%, respectively, and functional sensitivity of 4.0 ng/mL. The accuracy of the assays was verified by comparison with international reference standards, ensuring standardization across testing procedures. Vitamin D status was classified as deficient (25(OH)D < 10 ng/mL), insufficient (25(OH)D between 10 ng/mL and 29.9 ng/mL), or sufficient (25(OH)D greater than or equal to 30 ng/mL).

The variables of interest from complete blood count (hemoglobin level, MCV, MHC) were determined using an automatic analyzer based on flow cytometry with fluorescence. The serum iron level was measured using a spectrophotometric method. Intra- and inter-assay coefficients of variation were <5%. Hemoglobin and serum iron levels were categorized as normal status, iron deficiency, or iron-deficiency anemia. Iron deficiency was defined as serum iron levels < 50 mcg/dL) and iron-deficiency anemia was defined as iron deficiency and hemoglobin level below 2 standard deviations from the mean for age and gender.

### 2.3. Statistical Analysis

The statistical analysis included descriptive statistics (mean, standard deviation) as well as elements of inferential statistics. Correlations between BMI, vitamin D levels, and hematological parameters were analyzed using the Pearson correlation coefficient of linear regression after adjustment for relevant confounding variables (age and sex). A *p*-value of less than 0.05 was considered statically significant.

All parents/care-givers signed the informed consent for the children inclusion in the study.

## 3. Results

The cohort included 69 patients; the mean age was 11.33 ± 4.05 years (mean ± SD). Of these, 58% were female and 66.6% patients were from urban areas. The mean BMI value was 27.19 ± 6.40 kg/m^2^ (mean ± standard deviation-SD). According to the CDC2022, the patients had the following distribution: 37.7% overweight, 42% obese, and 20.3% severely obese children. Males had a higher risk for obesity in this cohort (48.3%), and females were more likely to be in the overweight or obesity category. Obesity was more prevalent in boys than in girls (48.3% vs. 37.5%), but there was no significant difference between the BMI status in relation to sex in this study (*p* = 0.178).

Regarding age categories, we noticed that pre-adolescents (10–13 years old) were more likely to be in the overweight or obesity category, with an equal distribution of 47.8%. The adolescent group (14–18 years old) had approximately the same distribution between the BMI categories. The risk of being obese or severely obese increases with the age of the patients.

The distribution of BMI status among the participants and their demographic characteristics are all summarized in Table 1. No statistically significant differences were observed between the BMI distribution in relation to sex, age group, or place of residence.

Table 2 presents a comparison of the mean values for the clinical characteristics and biochemical measurements for each BMI category (overweight, obese, and severely obese). No statistically significant differences were found in the analyzed variables across the different groups based on BMI status.

The mean 25(OH)D level was 26.83 ± 10.62 ng/mL (mean ± SD), considered an insufficient vitamin D status. Patients with severe obesity exhibited lower average levels of 25(OH) when compared with the other BMI categories (overweight and obesity), from mean values of 28.4 ng/mL in overweight to 24.8 ng/mL in children with severe obesity. However, 25(OH)D levels were not significantly higher in the overweight and obese groups (*p* = 0.575).

Mean serum iron levels were lower in the severe obesity category, but with no significant difference compared with the overweight or obese groups (*p* = 0.517).

Regarding calcium and phosphorus plasma levels, the data were incomplete for all 69 patients, considering the retrospective design of this study. However, based on the available data for some of the patients, there were no abnormalities in phospho-calcium metabolism, with values within the range of normal for this cohort.

Insufficient 25-hydroxyvitamin D levels were found in 66.6% of patients, with deficiency in 2.9%. Insufficiency rates were significantly higher in severely obese children (71.5%), with a lower prevalence in overweight (61.5%) and obese patients (69%). Figure 1 shows the prevalence of vitamin D insufficiency (25(OH)D levels < 30 ng/mL) in relation to the BMI classification.

We analyzed the association between the 25(OH) level and age for each patient using the Pearson correlation coefficient of linear regression. Figure 2 shows the result, with a moderate statistically significant correlation between age and vitamin D levels. The Pearson coefficient was r = −0.444 with a *p*-value < 0.0001, meaning that as children age, their vitamin D levels tend to decrease.

A moderate inverse correlation was found between vitamin D levels and BMI (Pearson coefficient r = −0.31, *p* = 0.015), indicating that a higher BMI is linked to lower levels of 25-hydroxyvitamin D. The result of the correlation is presented in Figure 3.

The mean serum iron level was 66.89 ± 23.03 mcg/dL (mean ± SD), considered as a normal value for this cohort age group. The prevalence of iron deficiency was assessed across BMI categories. Iron deficiency was identified in 36.2% of children. The prevalence of iron deficiency was higher among the overweight (50%) compared to obesity and severe obesity (24.2%, respectively 28.6%). No statistically significant correlation was found between serum iron levels and BMI in this cohort (Pearson correlation coefficient r = −0.02, *p* > 0.05). The distribution of serum iron levels according to BMI status is presented in Figure 4.

Iron-deficiency anemia was observed in 27.5% of patients. Similarly, iron-deficiency anemia was more prevalent in overweight children (38.5%) than in the obese (17.3%) and severely obese categories (28.6%). The results are shown in Figure 5.

## 4. Discussions

Serum 25(OH)D is not only a predictor of bone health, but it is also an independent predictor of other chronic diseases: hypertension, dyslipidemia, and type 2 diabetes [4,5,6,26,27]. Obesity significantly affects vitamin D status and metabolism, placing obese children and adolescents at a higher risk of vitamin D insufficiency or deficiency [28]. Considering these aspects, it is important to provide valuable data for developing early identification strategies and personalized treatment approaches for at-risk pediatric populations.

Sex, group age, and residence location have been examined as independent factors associated with the BMI categories (overweight, obese, and severely obese). A study from Romania analyzing demographic characteristics in obese children proved that the mean age of the children with obesity included was 10.25 ± 3.28 (mean± SD), with a mean BMI of 26.87 ± 4.83 kg/m^2^ [29]. In relation to our study, the findings align with those mentioned above, as we also observed a mean age of 11.33 ± 4.05 years and a mean BMI of 27.19 ± 6.40 kg/m^2^. Other anterior studies reported a higher mean age of 14.9 ± 1.4 years, suggesting that obesity is occurring at increasingly younger ages [30]

It was presented in several studies that girls, particularly adolescents, presented higher BMIs, with lower vitamin D levels compared to boys [31,32]. In our case, the analysis did not find a significant difference in the distribution of BMI status according to gender.

A study from the US discovered a higher prevalence of insufficient levels of vitamin D in severely obese children [9]. In relation to our study, the findings are comparable, as we also observed significantly higher rates of insufficient 25(OH)D levels in severely obese children (71.5%) compared to the overweight and obese groups. The severely obese population showed lower mean levels of 25(OH)D when compared with the other BMI groups (overweight and obesity), as indicated by the results of the logistic analysis in this study. The rates may vary among other published studies depending on race or geographic region [32,33].

A significant negative correlation was observed between the 25(OH)D level and age in children, as shown in a large study, with 25(OH)D levels decreasing by 0.17 ng/mL per month with age [34]. Our findings sustain this connection, since the cohort in our research also presented a negative correlation between vitamin D levels and age, suggesting that pre-adolescents have an increased risk of developing insufficient levels of 25(OH)D. This association can be caused by the fact that, as they grow, children are experiencing rapid periods of linear growth, increasing their needs and leading to deficiencies [35]. In addition, as they grow, adolescents are less compliant with oral vitamin D supplementation, as well as due to several behavioral factors (limited outdoor activities, increased screen time) [35].

This study, like several other studies, has demonstrated an inverse correlation between circulating 25-hydroxyvitamin D (25(OH)D), the main indicator of vitamin D status, and body mass index [36,37,38,39]. This association is largely attributed to the lipophilic nature of vitamin D, leading to its sequestration in fat tissue, thereby reducing its availability for biological functions [12,36,40].

Several studies compared the prevalence of iron deficiency across different weight groups [19,41]. In a study conducted on a large sample of 9698 children, the prevalence of iron deficiency was found to increase as BMI rose from normal weight to overweight children [19]. Furthermore, more findings suggested that iron deficiency was more prevalent in overweight boys and girls (60.0%) than in those with obesity or normal weight (50.0% and 31.8%) [42]. Our findings align with these data as our cohort also presented a more significant prevalence in overweight children for iron deficiency (50%) and iron-deficiency anemia (38.5%). Our study showed no significant differences between BMI and hematological status (r = −0.02). In contrast, there are medical data showing a negative correlation of low serum iron levels with BMI (r = −0.44, *p* < 0.001), with no association with gender or age [20].

Considering the rising prevalence of overweight children and the well-documented complications of iron deficiency, these findings indicate that screening guidelines for iron deficiency may need to be adjusted for children with increased BMI [16].

Several studies concluded that the significant association between overweight children and iron deficiency can be explained by a variety of factors, including genetic influences, poor diet with fewer iron-enriched foods, lack of physical activity, along with increased iron requirements, impaired iron absorption in obese individuals, and an increase in hepcidin concentration [21,43,44,45].

This study has a number of limitations, including the relatively small population, its retrospective observational design, and the limited information on physical activity, sun exposure, and sunscreen use. A detailed nutritional evaluation (covering dietary intake of vitamin D, daily supplementation, etc.) was not available considering the retrospective nature of the study, so these details were not factored into the statistical analysis, a limitation that may have impacted the results. In our experience, vitamin D supplementation is not a common practice, especially after the age of 2, potentially influencing the rate of vitamin D deficiency observed in this population.

Recommendations regarding a hypocaloric diet and a physical activity regimen were made for all patients in this cohort, and vitamin D and iron supplementation were initiated where required. The authors consider evaluating the patients’ outcomes with dietary/lifestyle changes and substitution therapy with vitamin D or/and oral iron as part of future research.

## 5. Conclusions

Vitamin D insufficiency is highly prevalent in severely obese children, and this study emphasizes the need for targeted screening, along with appropriate interventions such as dietary adjustments and vitamin D supplementation to avoid metabolic complications. A statistically significant negative correlation between BMI status and 25(OH)D levels was also demonstrated.

Additionally, the study found a high prevalence of iron deficiency and iron-deficiency anemia in the overweight category, suggesting that adipose tissue contributes to chronic inflammation, disrupting iron homeostasis. Interventions should target improving dietary quality, reducing inflammation, and ensuring appropriate iron supplementation when necessary.

The findings of this study enhance our understanding of the cause–effect–consequence relationships between obesity and nutritional deficiencies in children. Considering the widespread prevalence of nutritional deficiencies in this population, implementing systematic screening and treatment programs would be beneficial to prevent long-term adverse outcomes.

## Figures and Tables

**Figure 1 nutrients-17-01193-f001:**
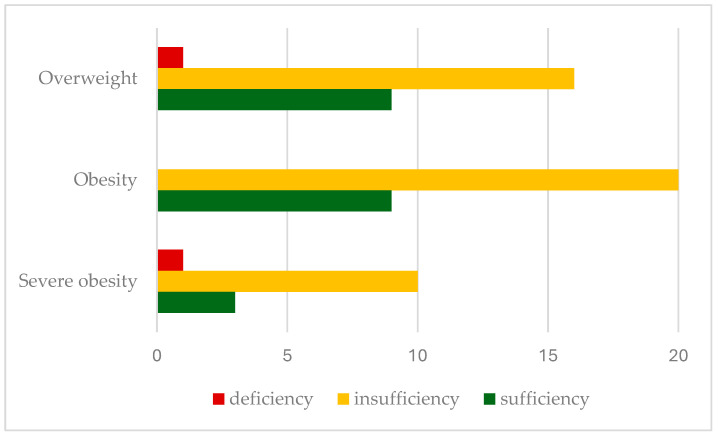
Prevalence of serum vitamin D status according to BMI classification.

**Figure 2 nutrients-17-01193-f002:**
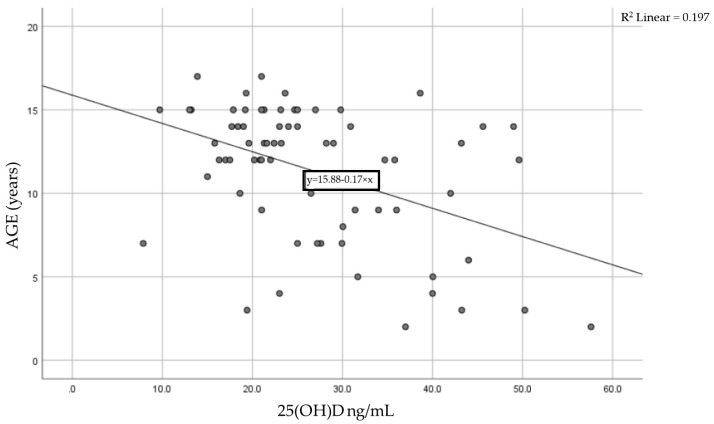
Correlation between age and serum vitamin D levels in the studied cohort.

**Figure 3 nutrients-17-01193-f003:**
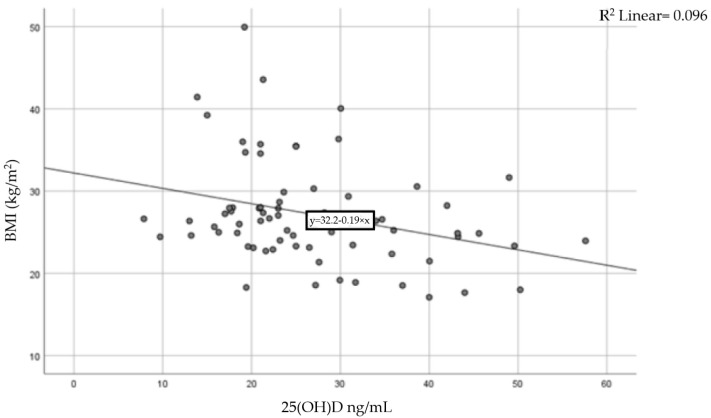
Correlation between BMI and serum vitamin D levels.

**Figure 4 nutrients-17-01193-f004:**
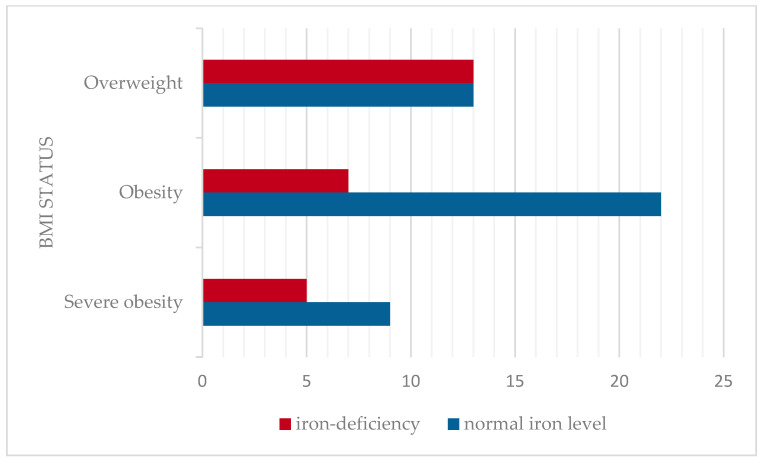
Prevalence of iron deficiency according to BMI classification.

**Figure 5 nutrients-17-01193-f005:**
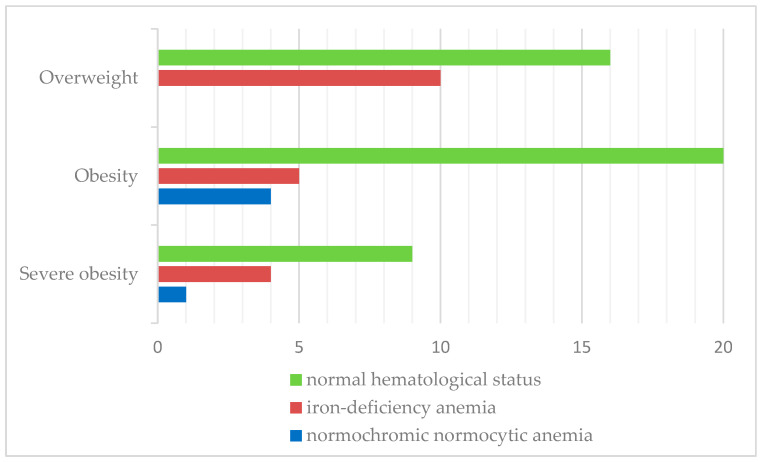
Distribution of hematological status according to BMI classification.

**Table 1 nutrients-17-01193-t001:** Demographics of the patients according to the BMI status.

	Overweight, n (%)	Obesity, n (%)	Severe Obesity, n (%)	*p* Value
**Sex**				*p* = 0.178
Male	10 (34.5%)	14 (48.3%)	5 (17.2%)	
Female	16 (40%)	15 (37.5%)	9 (22.5%)	
**Age group**				*p* = 0.178
2–5 years old	4 (44.4%)	3 (33.3%)	2 (22.2%)	
6–9 years old	3 (27.3%)	6 (54.5%)	2 (18.2%)	
10–13 years old	11 (47.8%)	11 (47.8%)	1 (4.3%)	
14–18 years old	8 (30.8%)	9 (34.6%)	9 (34.6%)	
**Residence**				*p* = 0.065
Rural	5 (21.7%)	14 (60.9%)	4 (17.4%)	
Urban	21 (45.7%)	15 (32.6%)	10 (21.7%)	

**Table 2 nutrients-17-01193-t002:** Clinical and biochemical characteristics based on the BMI status (mean ± SD).

	Overweight, n = 26	Obesity, n = 29	Severe Obesity, n = 14	*p* Value
Age	11.04 ± 4.09	11.24 ± 3.55	12.07 ± 5.06	*p* = 0.740
BMI	22.70 ± 2.97	27.02 ± 3.36	35.84 ± 7.22	
Hemoglobin (g/dL)	12.76 ± 1.55	13.30 ± 1.71	13.42 ± 1.35	*p* = 0.273
MCH (pg)	27.56 ± 3.18	27.39 ± 2.09	25.96 ± 2.62	*p* = 0.167
MCV (fL)	79.84 ± 5.66	80.38 ± 5.44	78.53 ± 6.43	*p* = 0.613
Serum iron level (mcg/dL)	70.58 ± 48.03	65.69 ± 29.08	56.71 ± 19.90	*p* = 0.517
Vitamin D (ng/mL)	28.40 ± 11.59	26.40 ± 8.62	24.80 ± 12.68	*p* = 0.575
Calcium level	9.79 ± 0.42	10.04 ± 0.33	10.04 ± 0.39	*p* = 0.092
Phosphorus	4.64 ± 0.83	5.42 ± 1.68	5.43 ± 1.00	*p* = 0.379

## Data Availability

The original contributions presented in the study are included in the article; further inquiries can be directed to the corresponding author.

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
