# Peer review of "Customizing Nutrients: Vitamin D and Iron Deficiencies in Overweight and Obese Children—Insights from a Romanian Study"

_nutrients, 2025, doi:10.3390/nu17071193_

Round 1
Reviewer 1 Report
Comments and Suggestions for Authors
Thank you for submitting the manuscript "Customising Nutrients: Vitamin D and Iron Deficiencies in Overweight and Obese Children – Insights from a Romanian Study" to Nutrients. The manuscript is well written, however, as the data collection was very specific to some biochemical tests, the discussion seems a bit superficial. It would be possible for the authors to expand on other results that may have been collected at the same time, such as ferritin, vitamin B12, etc. In addition, it would be important to separate gender groups, since there are specific phases in adolescence, such as a growth spurt that occurs at different ages for different genders.
- As this is an experimental research article, I suggest changing the structure of the introduction to a single body (without subitems) so that it is more integrated and readable.
- Figure 1 could be improved to contain fewer numbers on the x-axis and be "cleaner". All figures need to be modified so that the font is the same as in the text. - Figures 2 and 3 appear a little blurry. Could this be the resolution?
Comments on the Quality of English Language
English is fine
Author Response
For research article: Customising Nutrients: Vitamin D and Iron Deficiencies in Overweight and Obese Children – Insights from a Romanian Study (nutrients-3541034)
Response to Reviewer |
||
1. Summary |
|
|
Thank you very much for taking the time to review this manuscript. Please find the detailed responses below and the corresponding revisions and corrections highlighted in the re-submitted files. |
||
2. Point-by-point response to Comments and Suggestions for Authors |
||
Comments 1: Thank you for submitting the manuscript "Customising Nutrients: Vitamin D and Iron Deficiencies in Overweight and Obese Children – Insights from a Romanian Study" to Nutrients. The manuscript is well written, however, as the data collection was very specific to some biochemical tests, the discussion seems a bit superficial. It would be possible for the authors to expand on other results that may have been collected at the same time, such as ferritin, vitamin B12, etc. |
||
Response 1: Thank you for pointing this out. As mentioned in the Material and Method section, the collected data focused specifically on the analyses stated: “hemoglobin level (g/dL), mean corpuscular volume (MCV, fL) and mean corpuscular hemoglobin (MHC, pg). The analysis also included serum 25-hydroxyvitamin D (25(OH)D) level (ng/mL), serum iron level (mcg/dL), calcium and phosphorus plasma levels.” Given the retrospective nature of the study, other parameters such as ferritin and vitamin B12, as you suggested, were not collected or included. We discussed data results on vitamin D, serum iron levels and specified hematological parameters from the complete blood count to assess hematologic status and classify anemia. As stated in the Results section, page 5, line 182, we did not have complete data regarding serum calcium and phosphorus for all patients in the cohort, which did not allow to provide further details beyond those presented in Table 2 or to establish correlations without statistical errors: ”Regarding calcium and phosphorus plasma levels, the data were incomplete for all the 69 patients, given the retrospective nature of this study. However, based on the available data for some of the patients, there were no abnormalities in phospho-calcium metabolism, with values within the range of normal for this cohort.” Therefore, we did not include them in the Discussion section. |
||
Comments 2: In addition, it would be important to separate gender groups, since there are specific phases in adolescence, such as a growth spurt that occurs at different ages for different genders. |
||
Response 2: I agree with this comment. Puberty evaluation might have brought interesting details regarding gender disparities in growth, bone health and calcium and phosphate homeostasis. Unfortunately, clinical details on puberty onset were not available in the patient’s charts to be gathered retrospectively, so the authors could not factor gender specific growth spurts into the statistical analysis. Comments 3: As this is an experimental research article, I suggest changing the structure of the introduction to a single body (without subitems) so that it is more integrated and readable. Response 4: Agree. I have accordingly modified the structure of the introduction to a single body and eliminated the subitems. Comments 4: Figure 1 could be improved to contain fewer numbers on the x-axis and be "cleaner". All figures need to be modified so that the font is the same as in the text. Response 4: I have modified the x-axis of the Figure 1 and also, the font for all figures in the manuscript, according to your suggestion. Comments 5: Figures 2 and 3 appear a little blurry. Could this be the resolution? Response 5: It was an image resolution issue, but I fixed it. |
||
3. Response to Comments on the Quality of English Language |
||
Point 1: English is fine |
||
Response 1: The manuscript was proof-read and corrected by a native English speaker. |
Reviewer 2 Report
Comments and Suggestions for Authors
The study shows interesting results. To increase the quality of the paper, the following points might be reconsidered.
- In Abstract, the demographic of studied subjects must be described.
- In Abstract, the definition of overweight states must be shortly described.
- In Abstract, the definition of hypovitaminosis must be shortly described.
- In Abstract, the correlations must be concretely described.
- In Methods, this study is deemed to be a retrospective cohort study, while in Discussion, this study was described as a cross-sectional design. Which is right as these designs were different?
- Methods; inclusion/exclusion criteria should be described.
- In the part of criteria, could any medicine be included/excluded?
- Methods; the methods including fasting/non-fasting states in the blood collection could be detailed.
- Methods; the vitamin D-related markers (concentrations) are known across assays used. The assay performances (including company names) and accuracy/standardization should be described.
- Methods; similarly, assay information should be added regarding the iron markers used for the study.
- Methods; how did the subjects give an informed consent. It should be detailed.
- Results; SD can be fully spelled out at the first appearance, and then abbreviated.
- Table 2; the added expression of SD in obesity column of vitamin D would be a typo.
- How were the adjusted correlation analyses performed? Were the results fully demonstrated?
- Line 179; the unit of iron would be a typo.
- As the weak point of the study, the regulation of dietary factors was not done. How were the dietary patterns and components presumable for the studied subjects in your setting? Even though it is presumable, please discuss it more.
Comments on the Quality of English Language
Good. Please check it with native again.
Author Response
For research article: Customising Nutrients: Vitamin D and Iron Deficiencies in Overweight and Obese Children – Insights from a Romanian Study (nutrients-3541034)
Response to Reviewer |
||
1. Summary |
|
|
Thank you very much for taking the time to review this manuscript. Please find the detailed responses below and the corresponding revisions and corrections highlighted in the re-submitted files. |
||
2. Point-by-point response to Comments and Suggestions for Authors |
||
Comments 1: In Abstract, the demographic of studied subjects must be described. Response 1: Thank you for pointing this out. I have briefly described the demographic of studied subjects in the Abstract. You can find the changes in page 1, line 16. Comments 2: In Abstract, the definition of overweight states must be shortly described. Response 2: Agree. I have accordingly described the weight categories: overweight, obese and severely obese. You can find the changes in page 1, line 19. Comments 3: In Abstract, the definition of hypovitaminosis must be shortly described. Response 3: I agree, therefore I shortly described the definition of hypovitaminosis D in Abstract: page 1, line 22. Comments 4: In Abstract, the correlations must be concretely described. Response 4: Thank you for pointing this out. Therefore, I described the correlations mentioned in Abstract: page 1, lines 27, 28 and 29. Comments 5: In Methods, this study is deemed to be a retrospective cohort study, while in Discussion, this study was described as a cross-sectional design. Which is right as these designs were different? Response 5: Thank you for pointing out this mistake. This study is a retrospective observational study, as it is mentioned in Methods. I have modified this information in Discussion: page 9, line 28.2 Comments 6: Methods; inclusion/exclusion criteria should be described. Response 6: I have accordingly revised the inclusion/exclusion criteria for this cohort study and therefore inserted a paragraph with this information, on Methods, page 3, line 102: “The inclusion criteria consisted in age between 2 and 18 years, the status of being overweight or obese (based on age- and gender-specific BMI-percentiles provided by Center for Disease Control) and availability of required data. The exclusion criteria were: genetic or metabolic diseases, patients with incomplete data or those whose care-givers refused to sign the informed consent prior to the inclusion in the study.” Comments 7: In the part of criteria, could any medicine be included/excluded? Response 7: In the part of criteria, we did not include/exclude any medication for the children in this cohort. Comments 8: Methods; the methods including fasting/non-fasting states in the blood collection could be detailed. Response 8: Agree. I added this information in the manuscript and you can find it in Methods, page 3, line 120: “Blood samples were collected from all patients by venipuncture, in a fasting state (for 8 to 12 hours of fasting) since they were all admitted to hospital.” Comments 9: Methods; the vitamin D-related markers (concentrations) are known across assays used. The assay performances (including company names) and accuracy/standardization should be described. Response 9: I revised and added this additional information in the manuscript. This paragraph can be found in Methods, page 3, line 126: “The 25-hydroxyvitamin D was determined by a high-specific immunochemical method with detection by electrochemiluminescence with intra- and inter-assay coefficient of variation of 4.5% and 7.9%, respectively, and functional sensitivity of 4.0 ng/mL. The accuracy of the assays was verified by comparison with international reference standards, ensuring standardization across testing procedures.” Comments 10: Methods; similarly, assay information should be added regarding the iron markers used for the study. Response 10: I have modified the information to emphasize this point and added this paragraph in Methods, page 3, line 134: “The variables of interest from complete blood count (hemoglobin level, MCV, MHC) were determined using an automatic analyzer based on flowcytometry with fluorescence. The serum iron level was measured using a spectrophotometric method. Intra- and interassay coefficients of variation were <5%.” Comments 11: Methods; how did the subjects give an informed consent. It should be detailed. Response 11: I have completed this information in Methods, page 4, line 148 with this paragraph: “All parents/care-givers signed the informed consent for the children inclusion in the study. “ The study was performed according to the principles of the Helsinki Declaration and it was approved by the Ethics Committee of “Grigore Alexandrescu” Emergency Clinical Hospital for Children in Bucharest, Romania, no 40/05.11.2024. Comments 12: Results; SD can be fully spelled out at the first appearance, and then abbreviated. Response 12: Agree. I have modified and fully spelled “standard deviation” with the “SD” abbreviation as it’s the first appearance in text – page 4, line 153. Comments 13: Table 2; the added expression of SD in obesity column of vitamin D would be a typo. Response 13: Indeed, it was a typo and I corrected it in Table 2. Comments 14: How were the adjusted correlation analyses performed? Were the results fully demonstrated? Response 14: Correlations were adjusted for relevant confounding variables (age and sex). I revised and completed the information to emphasize this point in Methods, page 4, line 146. The adjusted correlation results were clearly presented in text (including Pearson correlation coefficients and p-values) and figures (Figure 2 and Figure 3). The inverse correlation between variables is suggested by the negative slope of the regression line in the figures (Figure 2 and Figure 3), and the coefficient of determination (R²), which appears in the right corner Figure 2 and Figure 3 explains the variability of this relationship between variables, adjusted for the mentioned confounding factors. Comments 15: Line 179; the unit of iron would be a typo. Response 15: Agree. I corrected the iron unit. Comments 16: As the weak point of the study, the regulation of dietary factors was not done. How were the dietary patterns and components presumable for the studied subjects in your setting? Even though it is presumable, please discuss it more. Response 16: In order to emphasize this point, I have revised the text and inserted additional information about this in Methods, page 3, line 115 : “Data collection included a dietary survey from the medical charts of the included patients. Due to the retrospective design of the study, this information was limited, but dietary errors were identified in all children, including: the consumption of processed foods, a large number of snacks between main meals and the consumption of calorie-dense but nutrient-poor foods.”. More than that, I have mentioned in Discussion, as a limitation of this study (page 9, line 283: ” A detailed nutritional evaluation (covering dietary intake of vitamin D, daily supple-mentation, etc.) was not available considering the retrospective nature of the study so these details were not factored into the statistical analysis, limitation that may have impacted the results.” |
||
3. Response to Comments on the Quality of English Language |
||
Point 1: Good. Please check it with native again. |
||
Response 1: The manuscript was proof-read and corrected by a native English speaker. |
Round 2
Reviewer 2 Report
Comments and Suggestions for Authors
The paper was improved. Although the correlation was shown in r-value in Abstract, it is suitable to show the value of age- and gender-adjusted correlation analysis. The exclusion criteria included the genetic or metabolic diseases. What disease types (names) were concretely excluded? How were the diseases diagnosed? Weren’t the overweight/obesity states based on the metabolic diseases? For the accuracy and reproducibility of the study, such information could be detailed.
Comments on the Quality of English Language
Mostly good.
Author Response
For research article: Customising Nutrients: Vitamin D and Iron Deficiencies in Overweight and Obese Children – Insights from a Romanian Study (nutrients-3541034)
Response to Reviewer |
||
1. Summary |
|
|
Thank you very much for taking the time to review this manuscript. Please find the detailed responses below and the corresponding revisions and corrections highlighted in the re-submitted files. |
||
2. Point-by-point response to Comments and Suggestions for Authors |
||
Comments 1: The paper was improved. Although the correlation was shown in r-value in Abstract, it is suitable to show the value of age- and gender-adjusted correlation analysis. |
||
Response 1: Thank you for pointing this out. The correlation results were clearly presented in text (including Pearson correlation coefficients values and p-values to indicate statistical significance.) and figures (Figure 2 and Figure 3). The values presented are obtained after adjusting for age and gender, resulting the partial correlation coefficients (r) and p values. We did not run a multivariable regression model, so we could not see the separate coefficients for each predictor. The inverse correlation between variables is suggested by the negative slope of the regression line in the figures (Figure 2 and Figure 3), and the coefficient of determination (R²), which appears in the right corner of Figure 2 and Figure 3 explains the variability of this relationship between variables. |
||
Comments 2: The exclusion criteria included the genetic or metabolic diseases. What disease types (names) were concretely excluded? How were the diseases diagnosed? Weren’t the overweight/obesity states based on the metabolic diseases? For the accuracy and reproducibility of the study, such information could be detailed. |
||
Response 2: I agree with this comment. We concretely excluded the following genetic syndromes: Down Syndrome, Prader-Willi Syndrome, Bardet-Biedl Syndrome, Beckwith-Wiedemann Syndrome. The authors clinically excluded these diseases based on lack of associated congenital malformations, specific phenotypic features, craniofacial dysmorphism or growth retardation. I have updated the manuscript and added these syndromes in the exclusion criteria. You can find the changes in Methods, page 3, line 105-106. Thank you for pointing out the comment about metabolic disease. To clarify this aspect, I have modified this statement in the manuscript with “inborn errors of metabolism, excluded using data in the patients’ medical records” (such as congenital lipodystrophies, galactosemia, disruptions in energy metabolism or hormone metabolism) (Methods, page 3, line 106-107). We did not investigate metabolic conditions associated with overweight/obesity status in the patients from the cohort (such as glucose metabolism disorders, dyslipidemia, hyperuricemia, or metabolic syndrome) because this was not the aim of the study, but it is an interesting direction for future research. |
||
3. Response to Comments on the Quality of English Language |
||
Point 1: Mostly good. |
||
Response 1: The manuscript was proof-read and corrected by a native English speaker. |